# Serum Osteocalcin, Sclerostin and Lipocalin-2 Levels in Adolescent Boys with Obesity over a 12-Week Sprint Interval Training

**DOI:** 10.3390/children10050850

**Published:** 2023-05-08

**Authors:** Marit Salus, Vallo Tillmann, Liina Remmel, Eve Unt, Evelin Mäestu, Ülle Parm, Agnes Mägi, Maie Tali, Jaak Jürimäe

**Affiliations:** 1Institute of Sports Sciences and Physiotherapy, Faculty of Medicine, University of Tartu, Ujula 4, 51008 Tartu, Estonia; liina.remmel@ut.ee (L.R.); evelin.maestu@ut.ee (E.M.); jaak.jurimae@ut.ee (J.J.); 2Department of Physiotherapy and Environmental Health, Tartu Health Care College, Nooruse 5, 50411 Tartu, Estonia; ylleparm@nooruse.ee; 3Department of Pediatrics, Institute of Clinical Medicine, Faculty of Medicine, University of Tartu, Lunini 6, 50406 Tartu, Estonia; vallo.tillmann@kliinikum.ee; 4Children’s Clinic of Tartu University Hospital, Lunini 6, 50406 Tartu, Estonia; 5Department of Sports Medicine and Rehabilitation, Institute of Clinical Medicine, Faculty of Medicine, University of Tartu, Puusepa 1a, 50406 Tartu, Estonia; eve.unt@kliinikum.ee (E.U.); maie.tali@kliinikum.ee (M.T.); 6Sports Medicine and Rehabilitation Clinic, Tartu University Hospital, Puusepa 1a, 50406 Tartu, Estonia; agnes.magi@kliinikum.ee

**Keywords:** osteokines, sprint interval training, bone mineral density, obesity, youth

## Abstract

The aim of the study was to examine the effects of supervised cycling sprint interval training (SIT) on serum osteocalcin, lipocalin-2 and sclerostin levels, and bone mineral characteristics among obese adolescent boys. Untrained obese adolescent boys aged 13.4 ± 0.3 were assigned to either a 12-week SIT group (3 sessions/week), or a non-exercising control group who continued with their habitual everyday life. Serum osteocalcin, lipocalin-2 and sclerostin concentrations, and bone mineral values were assessed before and after intervention. After 12-week intervention, where 14 boys in both groups ended the study, there were no significant differences in serum osteokine levels between the groups after 12 weeks, while whole body bone mineral content and lower limb bone mineral density increased in the SIT group (*p* < 0.05). Change in body mass index was negatively correlated with the change in osteocalcin (r = −0.57; *p* = 0.034), and positively correlated with the change in lipocalin-2 levels (r = 0.57; *p* = 0.035) in the SIT group. Supervised 12-week SIT intervention improved bone mineral characteristics, but did not change osteocalcin, lipocalin-2 or sclerostin levels in adolescent boys with obesity.

## 1. Introduction

The early detection of obesity-related health problems during childhood and adolescence is essential because the majority of them are preventable through changing risk behaviors such as unhealthy diet and/or low levels of physical activity [1]. Diet is a very important factor in weight management [2,3], but in conjunction with diet modification, exercise training is another major factor contributing to the treatment of overweight and obese individuals [4,5]. Exercise training is an effective strategy in decreasing excessive adiposity, together with the risk for obesity-related diseases, including type 2 diabetes [6] and different cardiovascular diseases [7,8]. Moderate-intensity continuous training (MICT) has mostly been used to reduce body fat mass and improve different health-related variables in young individuals with obesity [8,9]. However, high-intensity interval training (HIIT) is a good substitute to classical MICT programs, with higher compliance rates among overweight and obese individuals [10]. It appears that in contrast to MICT, HIIT protocols characterized by the variety of exercise intensity and short duration exercise bouts are more enjoyable for obese youth [11]. Furthermore, specific HIIT protocols may provide superior improvements in body mass, insulin sensitivity and cardiorespiratory fitness, along with lower ratings of perceived exertion compared to MICT, in overweight and obese populations [6,12]. Next to HIIT, sprint interval training (SIT) with shorter interval bouts is suggested to reduce body fat mass for young individuals with obesity [13,14]. The time efficiency of the SIT protocol is well characterized by the fact that only a 10 min per exercise session (3 × 20 s all-out cycle bouts with 2 min rest period) three times a week for 12 weeks was needed to improve indices of cardiometabolic health in sedentary individuals [7]. As mentioned, short sprint bouts are more enjoyable among obese children as this kind of training is more natural to childlike behavior full of short maximal sprints [15]. Therefore, with relatively higher exercise intensity but with less exercise volume, SIT is a time-efficient training strategy for obese adolescents. A meta-analysis of HIIT intervention studies among youth revealed that from the 17 studies included, only three were conducted with obese boys and none of the protocols consisted of an all-out SIT approach [16]. This knowledge gives a strong impetus to further investigate the effect of SIT on the different health indicators of obese youth. While a commonly used SIT protocol (short cycle sprints up to 30 s at a training intensity of ≥90% VO_2_max with active rest between bouts) is beneficial in improving cardiometabolic health indices in adult populations [17], no studies using this specific SIT protocol have been conducted in adolescent boys with obesity.

Previous studies have examined the role of different muscle-derived factors in regulating several adipose tissue adaptations to exercise training and it was found that there exists a hormonal regulatory loop between the muscle and adipose tissue [18]. However, less is known about the bone-derived factors and their effect in coordinating the response of adipose tissue to exercise training [19,20]. It has been suggested that bone tissue also exerts endocrine regulation and secretes various systemic humoral factors like osteocalcin, sclerostin and lipocalin-2, that are involved in regulating energy metabolism and adiposity in humans [18,21,22]. Carboxylated osteocalcin, expressed by active osteoblasts [21], acts as a hormone in glucose metabolism, increasing insulin secretion, sensitivity and energy expenditure. It is also suggested that carboxylated osteocalcin participates in bone formation and, therefore, is used as a bone formation biomarker [18]. In overweight children, serum osteocalcin levels are lower compared to non-overweight peers, and are inversely correlated with body mass index and insulin [23]. For example, in young males with obesity, a reduction of body fat after supervised MICT increased the level of osteocalcin [9]. Another osteokine, sclerostin, a glucoprotein secreted mainly by osteocytes is also involved in a negative regulation of glucose metabolism by upregulating adiposity [18,22]. Although Kurgan et al. [24] and Luziani et al. [25] suggested a role of this osteokine in response to training among obese individuals, no studies have included the SIT training modality among adolescent boys with obesity. Lipocalin-2, expressed by osteoblasts, has been implicated in regulating appetite and energy metabolism [26]. Circulating lipocalin-2 levels are increased in obese individuals. Moreover, lipocalin-2 is related to the buildout of obesity-related metabolic diseases [21,22,27]. Although, it has been found that a 12-week HIIT period lowered circulating lipocalin-2 levels together with decreased body fat values in young men with obesity [28], such findings among adolescent boys with obesity are still lacking. 

To the best of our knowledge, the benefits of SIT on specific bone-derived factors, including osteocalcin, sclerostin and lipocalin-2 have not yet been investigated in obese adolescent boys. The aim of the study was to determine the effects of 12-week supervised SIT on serum osteocalcin, lipocalin-2 and sclerostin levels, bone mineral density (BMD) and bone mineral content (BMC) characteristics in adolescent boys with obesity. Accordingly, the current study hypothesized that 12-week SIT leads to the increase of serum osteocalcin and to decreases of sclerostin and lipocalin-2 levels, together with an improvement in bone mineral values in obese adolescent boys.

## 2. Methods

### 2.1. Participants

Thirty-seven physically inactive adolescent boys with obesity (E66.0 by ICD-10) entered into the study. All participants had to meet the inclusion criteria: (a) body mass index (BMI) at or above the 95th percentile on the Estonian BMI chart; (b) healthy and inactive (participation only in mandatory physical education classes in school); and (c) no current use of prescribed medications that may influence metabolism or body composition. Individuals with a history of any acute or chronic medical conditions like cardiovascular disease or metabolic syndrome were excluded from the study. Participants were randomly placed into SIT (*n* = 18) and untrained control (CONT; *n* = 19) groups. Experimental procedures, the study process and aims were described to each participant and the participants’ legal guardian, and informed written consent was obtained from both. The present study was approved by the Medical Ethics Committee of the University of Tartu, Estonia (#282/T-5), which is in accordance with the ethical standards of the Helsinki Declaration. The participants were instructed to maintain their usual behavioral and eating habits throughout the entire study period. The effects of 12-week supervised SIT on cardiorespiratory fitness, fat mass, body fat % [29] and different cardiometabolic biomarkers [30] have been previously reported in this group of subjects. However, the effects of 12-week supervised SIT on serum osteocalcin, lipocalin-2, sclerostin levels, and bone mineral density and content have not been investigated yet. Therefore, in this paper we focus on the effects of 12-week supervised SIT on bone health, including different osteokines. 

### 2.2. Study Design and Training Protocol

At first, baseline measurements that included the assessment of fasting blood biochemical characteristics and bone mineral parameters were performed for all participants. After a 12-week intervention period, the same measurements were repeated. The SIT group performed 30 s sprints on an electronically-braked cycle ergometer (Wattbike Pro/Trainer, Vermont House, Willford Ind Est, Nottingham, UK) for 12 consecutive weeks (Monday, Wednesday, Friday) under the supervision of an experienced researcher (Table 1). Each SIT training session involved a warm-up period of 10 min and an active cool-down period of 5 min cycling at a low cadence (<50 rpm) against a light resistance (<50 W), in accordance with previous studies in obese adolescents [3,31,32]. The CONT group maintained their habitual lifestyle and did not intend to do any strenuous physical exercise [28], except mandatory physical education lessons performed twice weekly in the school environment. 

### 2.3. Anthropometric and Bone Mineral Measurements

Anthropometric measures included body height (Martin metal anthropometer, Switzerland) and body mass (A&D Instruments Ltd., Abingdon, UK). BMI was also calculated (kg.m^−2^). Whole body and lower body region (WB BMD and lower limbs BMD, g.cm^−2^), bone mineral density and bone mineral content for the whole body (WB BMC, g) were assessed via dual-energy X-ray absorptiometry (DXA, Hologic Discovery QDR Series, Waltham, MA, USA) [29,30]. 

### 2.4. Blood Sampling and Biochemical Analysis 

A 10-mL fasting blood sample from the antecubital vein was obtained in the morning. Serum was separated and then frozen at −80° for further analysis. Serum osteocalcin (ng·mL^−1^) was analyzed using Immulite 2000 (DPC, Los Angeles, CA, USA). The intra- and inter-assay coefficient of variations (CVs) for osteocalcin was less than 7%. Lipocalin-2 (ng·mL^−1^) was assessed by a commercially available enzyme-linked immunosorbent assay (ELISA) kit (R&D Systems Inc., Minneapolis, MN, USA). This assay had intra- and inter-assay CVs 3.1% and 6.1%, respectively, and the least detection limit was 0.4 ng·mL^−1^. Sclerostin (pg·mL^−1^) was determined using ELISA kit (R&D Systems Inc., Minneapolis, MN, USA) with a minimum detectable level of 1.7 pg·mL^−1^, an intra-assay CV 1.9% and an inter-assay CV 9.5%.

### 2.5. Statistical Analysis

Data are shown as mean ± standard errors (±SE). Normality was assessed with the Shapiro-Wilks method; and to examine the homogeneity of variance, Levene’s test was performed. An independent *t* test was used to determine differences between the SIT and CONT groups at baseline. In addition, to determine possible changes in bone-derived markers between the before and after intervention independently for both groups, a paired *t* test was used. Effect sizes (ES) were calculated using Cohen’s d, and the values were interpreted as follows: >0.8, large effect; 0.5, medium effect; <0.2 small effect. To assess associations between changes of osteokines and clinical variable such as BMI, in the course of 12-week SIT training period, Pearson correlation was used. Changes were calculated by subtracting the final value at 12 weeks from the corresponding initial value at baseline. The effects of the intervention on blood biochemical markers and body composition were studied using a one-way ANCOVA applied to one factor, including the group as a fixed factor (SIT, CONT), change during the before intervention to the after intervention interval as a dependent variable, and before intervention values of the variable studied as a covariate to better control for between-group differences. Pubertal stage was also included in the ANCOVA model as a covariate, but as there was observed only a slight difference in the effects of interest between the analysis models including/not including pubertal stage, it is not considered in the final ANCOVA model. As this was a secondary analysis from a parent study, a priori power analysis was not possible. Based on the primary outcome of VO_2_ peak resulting from sprint interval training on aerobic capacity in the parent study, a systematic review of the literature and meta-analysis [33], we estimated a sample size of at least 12 for the SIT group. Statistical analysis was conducted using SPSS statistical software (SPSS version 21.0 for Windows, SPSS Inc., Chicago, IL, USA). The significance level was set at *p* < 0.05.

## 3. Results

### 3.1. Anthropometric Characteristics 

After the 12-week intervention period, 14 participants finished in the SIT group (mean age 13.1 ± 0.4 yrs) and 14 participants in the CONT group (mean age 13.7 ± 0.4 yrs). Drop-out subjects discontinued intervention due to lack of interest (n = 4), inability to contact for post testing (n = 2), or the participation rate of training was lower than 70% (n = 3). Although the drop-out rate was substantial (24%), the drop-out subjects were similar in their body composition to the subjects who completed the intervention. None of the participants reported SIT-related injuries. The main anthropometric characteristics are shown in Table 2. Baseline anthropometric parameters were not different between the groups (*p* > 0.05). Body height significantly increased in both groups compared to baseline values (*p* < 0.001). While BMI was not modified at Before vs. After, body mass (*p* = 0.002) significantly increased after the study period in the CONT group. 

### 3.2. Bone Mineral and Bone Biochemical Parameters 

Bone mineral and bone biochemical parameters are presented in Table 2. Compared to baseline, WB BMC increased after the 12-week period in both the SIT and CONT groups (*p* = 0.005, and *p* = 0.005, respectively), while in the SIT group a significant increase with almost a large practical effect was seen in lower limb BMD (*p* = 0.029, ES = 0.75). ANCOVA results revealed no significant differences in any aforementioned parameters between the two groups over time. Although the mean change in osteocalcin showed a smaller reduction in the SIT compared to the CONT group, none of them were significant within or between the groups.

### 3.3. Correlations among Independent Variables

After the 12-week SIT period, the change in osteocalcin correlated negatively and the change in lipocalin-2 correlated positively with the change in BMI (r = −0.57, *p* = 0.034; r = 0.57, *p* = 0.035, respectively). No associations between the changes of sclerostin and the changes of BMI or other bone measurements were observed in either group.

## 4. Discussion

Sprint interval training is effective in enhancing cardiorespiratory fitness, body composition and other cardiovascular risk factors in obese adolescent boys [8,34]. Nevertheless, to the best of our knowledge, this is the first study to assess the effects of all-out SIT on bone mineral and bone-derived biochemical parameters among obese adolescent boys. This study demonstrated that the 12 weeks of SIT improved bone mineral values without changes in serum osteokine levels in obese adolescent boys. 

Several studies evaluating the effect of interval training on bone-derived markers have shown a positive effect [9,28], while others have demonstrated a negative or no effect of exercise training [35,36] among overweight and obese adults. In the current study, none of the measured osteokines showed any favorable within- or between-group change after the SIT study period. However, opposite trends, although not significant, were seen in serum osteocalcin levels in this study. One possible explanation might be the age of our subjects, during which the variation in serum osteocalcin levels starts to increase, and many our subjects might have reached the maturity when osteocalcin levels begin to decline [37]. We still observed that the decrease of the osteocalcin level in the SIT group was smaller compared to the CONT group (−1.22 ± 2.6 ng·mL^−1^ vs. −3.95 ± 2.6 ng·mL^−1^, respectively) showing possible effects of the 12-week SIT training on serum osteocalcin levels. These findings are similar to Josse et al. [4], where no change in total osteocalcin and sclerostin levels was seen after 12-week aerobic and resistance training among obese adolescent girls with overweight and obesity. In contrast, a significant decrease in serum osteocalcin was observed after 8-weeks of HIIT (4 × 4 min running/walking at 85–95% HR_max_ interspersed with 3-min active rest) among obese young females [38]. It appears that undercarboxylated osteocalcin is a biologically more active form related to energy expenditure, and therefore it should be considered to measure in addition to total osteocalcin [39]. Accordingly, 8-week MICT exercise program showed a significant increase in undercarboxylated osteocalcin level in young obese males, whereas total osteocalcin levels showed downward trend [9]. Additionally, the serum level of osteocalcin is significantly lower in subjects with obesity as compared to lean peers [23,40,41]. This finding is supported by the results from the current study, where a negative association between change in osteocalcin and change in BMI values was found in the SIT group. Similar association was observed in previous studies [23,42]—the lower the osteocalcin level, the higher BMI and fat mass (both for kg and %) and vice versa. The observed correlation between osteocalcin and BMI confirms the fact that osteocalcin affects fat metabolism [43], and that this bone-derived hormone is involved not only in the bone formation process, but also plays a role in the fat amount already in puberty. Reduction of adiposity can positively influence bone metabolism by increasing osteocalcin levels as well as improving body glucose homeostasis by better insulin secretion and sensitivity [43].

Although the change in serum lipocalin-2 concentration between the groups was not significant, to our best knowledge, this is the first study investigating possible changes in this bone-derived marker among obese adolescent boys after specific SIT protocol. Nazari et al. [44] found that a rope jump exercise protocol practiced for 8 weeks (3 days/week) among obese adolescent boys resulted in significant within-group decreases in body weight and BMI together with a decrease in serum lipocalin-2 levels. Results revealed a significant between-group difference in all aforementioned obesity characteristics but not in lipocalin-2 level between the rope jump and control groups. In our study, changes in body weight and BMI or lipocalin-2 levels within the groups did not reach the statistical significance level. Present study results are not in line with Nazari et al. [44], and the reasons for the differences in the results of these two studies may be the personal variations of subjects, controlled or uncontrolled dietary intake, differences in intervention protocols or the magnitude of the change in different adiposity characteristics. Based on that, it can be hypothesized that a decrease in body weight and BMI in the present study could have led to a significant decrease in lipocalin-2 levels in obese adolescent boys. A very interesting finding of our study was a significant correlation between change in BMI and change in lipocalin-2. It has been shown that lipocalin-2 levels are higher in obese compared to lean individuals [27], and that there lies a positive relationship between serum lipocalin-2 levels and the variables of obesity such as body weight and BMI [27,45], similarly to our study where the decrease in lipocalin-2 concentration was associated with a decrease in BMI. In individuals with excessive body fat mass, lipocalin-2 is an inflammatory marker produced by adipocytes and being so as a mediator for chronic low-grade inflammation, which subsequently disturbs insulin signaling and contributes to the development of insulin resistance and type 2 diabetes [46]. Findings also support a positive association of serum lipocalin-2 with other cardiovascular disease risk factors such as abnormal lipids profile or increased fasting glucose and insulin levels, after 12-week supervised HIIT intervention among young men with obesity [28].

It is reported that cardio-type exercise training at an intensity of 85–90% HR_max_ with a frequency of 3 times per week with a minimum duration of 30 min performed at least 12 weeks decreases sclerostin levels in adults [25]. We have previously reported that serum sclerostin concentration is positively associated with body fat values in children [19]. In the current study, sclerostin levels stayed unchanged together with no changes in body weight and BMI values in the SIT group. It has been found that the change of sclerostin depends on the training modality—running seems to have more positive impact on the concentration of sclerostin than cycling, done with the same amount of time and intensity, both in young males [47,48] and young females [49]. Nevertheless, results concerning the effect of exercise training, especially specific high-intensity intermittent training among young population, are still incomplete [25]. In addition to osteocalcin and lipocalin-2, serum sclerostin levels are also correlated with fasting glucose, insulin resistance and with several inflammatory and metabolic conditions among obese youth [50,51]. Therefore, it is crucial to continue to search for such associations between osteokines and other indices of adiposity, not just BMI, as it may abate cardiovascular disease risk factors and obesity-related metabolic disorders in adolescents with obesity.

It is reported that children with obesity due to their greater mechanical load on bones have a significantly higher bone mass compared to lean peers as estimated by BMD and BMC [52]. Current study results confirm it, as the increase in WB BMC was seen in both obese groups after the intervention. Secondly, it is known that physical exercise during adolescence has an anabolic effect on bone tissue and therefore increases BMD and BMC during teenage years [52]. The increase in lower limb BMD, as seen in the SIT group but not in the CONT group, after the intervention period in this study affirms the positive effect of exercise training on bone tissue. These findings are in line with a study among obese adolescent girls and boys, where 16-weeks of 30 s of cycling sprints at 75–90% VO_2_peak interspersed with 30 s active rest improved BMD and BMC in the whole body, lumbar spine and the hip region [5]. Klentrou and Kuovelioti [53] suggested that the response of bone turnover to training is heavily reliant on the exercise mode, duration and intensity, along with the maturity and the gender of participants. Although excessive body mass, particularly in youth, may compromise the skeleton by hormonal regulation pathways [41], SIT done as cycling sprints may contribute to higher lower limb BMD as seen in the current study. One of the obesity-related morbidities is different musculo-skeletal disorders [54]. The major benefit of implementing cycling SIT among obese populations is the reduction of increased mechanical stress on the body during joint movements. Therefore, currently used SIT protocols can be an effective training modality for improving overall bone health among obese adolescents without straining the joints, and therefore, may have a crucial role not only in the prevention of cardiovascular disease risk but also osteoporotic-related fractures in later life [53]. 

The current study has some limitations that should be noted when interpreting the results. First, the post-assessment of the dietary intake was insufficient and therefore, the analysis concerning dairy product intake and its associations with circulating levels of osteokines could not be carried out. Second, the participants did not receive any nutritional counselling during the study. It has been shown that diet intervention or nutritional education during the training period plays an important role in regulating osteokine values [4] and thus produces better results in body weight regulation and in other cardiovascular risk factors [5] in obese individuals. Another potential limitation was the measurement of total osteocalcin, and not undercarboxylated form of osteocalcin, in serum. The latter is a more biologically active form related to energy expenditure [39]. Moreover, a relatively small sample size can be considered as a limitation of the study, but it is still comparable with previous studies with similar sample sizes [13,24,28,44]. One methodological limitation of this study may be that the warm-up period was too long, which may have a possible effect on the final results. Nevertheless, the study cohort was made up only of boys in a similar age range, and therefore reflects the situation among a male adolescent cohort. 

## 5. Conclusions

In obese adolescent boys who went through the supervised 12-week SIT intervention, serum levels of osteocalcin, lipocalin-2 and sclerostin did not change significantly, although bone mineral characteristics were significantly increased. The change in BMI was negatively correlated with the change in osteocalcin and positively with the change in lipocalin-2 levels. These results give us a new insight into how sprint interval training may affect the growing skeleton of obese adolescent boys.

## Figures and Tables

**Table 1 children-10-00850-t001:** Sprint interval training.

Variable	Week 1–4	Week 5–8	Week 9–12
Warm-up (min)	10	10	10
Sprint intervals	4	5	6
Interval time (s)	30	30	30
Recovery time between intervals (min)	4	4	4
Cool-down (min)	5	5	5
HR_max_ (%)	81.7	82.5	79.8
Peak power (W/kg)	5.8	6.5 *	7.0 *^#^
Attendance (%)	89.0	81.0	80.0
Workload time per session (min)	2.0	2.5	3.0
Training time per session (min)	29.0	33.5	38.0
Weekly training time (min)	87.0	100.5	114.0

HR_max_, maximum heart rate. * Significantly different (*p* < 0.05) from the corresponding values of Week 1–4; ^#^ Significantly different (*p* < 0.05) from the corresponding values of Week 5–8.

**Table 2 children-10-00850-t002:** Mean (±SE) clinical characteristics in sprint interval training (SIT) and control (CONT) groups before and after 12-week study period.

Variable	SIT (*n* = 14)			CONT (*n* = 14)			Difference ^a^ (95% CI)		
	Before	After	ES	Before	After	ES	SIT	CONT	*p*
Height (cm)	170.6 ± 2.7	172.7 ± 2.6 ^#^	1.77	173.5 ± 2.9	174.8 ± 2.8 ^#^	1.50	2.0 ± 0.3 (1.4, 2.5)	1.5 ± 0.3 (1.0, 2.1)	0.216
Body mass (kg)	89.1 ± 4.3	90.3 ± 4.1	0.46	99.3 ± 6.4	102.0 ± 6.5 ^#^	1.05	1.2 ± 0.7 (−0.3, 2.7)	2.8 ± 0.7 (1.3, 4.3)	0.141
BMI (kg·m^−2^)	30.3 ± 0.9	30.0 ± 0.9	0.32	32.6 ± 1.6	33.0 ± 1.6	0.39	−0.3 ± 0.3 (−0.8, 0.3)	0.4 ± 0.3 (0.1, 1.0)	0.088
WB BMD (g·cm^2^)	1.05 ± 0.03	1.06 ± 0.03	0.33	1.15 ± 0.05	1.13 ± 0.04	0.13	−0.01 ± 0.02 (−0.06, 0.04)	0.003 ± 0.02 (−0.05, 0.05)	0.645
LL BMD (g·cm^2^)	1.16 ± 0.04	1.19 ± 0.03 ^#^	0.75	1.29 ± 0.05	1.28 ± 0.05	0.13	0.009 ± 0.03 (−0.05, 0.07)	0.002 ± 0.03 (−0.06, 0.06)	0.870
WB BMC (g)	2247.98 ± 123.0	2325.05 ± 123.0 ^#^	0.89	2478.63 ± 160.2	2560.89 ± 170.6 ^#^	0.90	80.10 ± 24.4 (29.83, 130.4)	79.23 ± 24.4 (28.96, 129.5)	0.980
Osteocalcin (ng·mL^−1^)	26.31 ± 4.9	24.66 ± 3.6	0.12	24.34 ± 4.3	20.84 ± 3.6	0.31	−1.22 ± 2.6 (−6.63, 4.20)	−3.95 ± 2.6 (−9.36, 1.46)	0.241
Sclerostin (pg·mL^−1^)	275.18 ± 30.5	284.97 ± 27.6	0.24	284.49 ± 24.4	287.33 ± 26.8	0.05	9.17 ± 13.3 (−18.14, 36.48)	3.46 ± 13.3 (−23.85, 30.77)	0.764
Lipocalin-2 (ng·mL^−1^)	33.07 ± 1.9	31.00 ± 1.7	0.32	37.46 ± 3.2	33.67 ± 1.6	0.32	−3.77 ± 1.6 (−7.05, −0.48)	−2.09 ± 1.6 (−5.38, 1.20)	0.470

SE, standard error; CI, confidence interval; ES, effect size; BMI, body mass index; WB BMD, whole body bone mineral density; LL BMD, lower limb bone mineral density; BMC, bone mineral content; ^a^ Group differences examined by ANCOVA; ^#^ Significant differences (*p* < 0.05) from baseline.

## Data Availability

The datasets used in this study are available from the corresponding author on reasonable request.

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
