# Peer review of "Serum Osteocalcin, Sclerostin and Lipocalin-2 Levels in Adolescent Boys with Obesity over a 12-Week Sprint Interval Training"

_children, 2023, doi:10.3390/children10050850_

Round 1

Reviewer 1 Report (New Reviewer)

The reviewer would like to thank the authors for the submission of this work.

Investigating metabolic changes in response to exercise interventions in children with obesity is of high interest. However, a few questions remain after having reviewed the manuscript.

1. Why was age not adjusted for? It appears that WB BMC trended upward, but since it applies to both groups likely as a function of maturity. Specifically in children, it is important to consider the maturation factor when assessing outcomes. Please clarify.

2a. Why was uncarboxylated osteocalcin not assessed in the first place? From a physiological rationale, it is the uncarboxylated osteocalcin that should be of interest as a proxy-marker to increased bone remodeling in response to exercise.

2b. The introduction briefly mentions carboxylated osteocalcin but refrains from further detail. Please insert adequate background.

3. The results are currently suggesting an overall negative correlation between changes in BMI and changes in osteocalcin, and overall positive correlation with BMI changes and changes in lipocalin-2. However, in the abstract, you describe these changes only to be seen in the SIT group. Please correct.

4. Please omit the "salami-slicing" sentence from the end of your introduction. It appears that breaking-up outcome reports in smaller papers is doing exactly that. No need to elaborate beyond line 113 on this.

5. LL-BMD Confidence intervals for group differences both cross 0. This indicates a level of uncertainty and non-significance.

Author Response

Reviewer 2 Report (New Reviewer)

Dear Authors,

It was a pleasure to review your paper. I have attached a PDF copy of your paper with comments. I do require the following changes:

1 You need to acknowledge that exercise science in conjunction with diet modification can be effective in reducing adipose tissue. Throughout the introduction, the fact is not acknowledged. 

2. How an adequate sample size for the study was determined needs to be noted in the methodology.

3. A serious concern for me is that the 10-minute warmup has not been described. The warmup was significantly longer than the intense component (sprinting), therefore you need to detail the warmup. 

4. In the limitations you also need to acknowledge the possible effect of the warmup as well. 

5. Due to the limitations of the study, you need to soften the language used in the conclusion. 

Author Response

Reviewer 3 Report (Previous Reviewer 3)

This is the first study to assess the effects of SIT on bone mineral and bone-derived biochemical parameters among obese adolescent boys. This study showed that the 12 weeks of SIT improved bone mineral values without affecting serum osteokine levels in obese adolescent.

 The study aims to investigate the effects of SIT on bone-derived factors, such as osteocalcin, sclerostin, and lipocalin-2, in adolescent boys with obesity. The hypothesis is that 12 weeks of supervised SIT will increase serum osteocalcin levels and decrease sclerostin and lipocalin-2 levels, as well as improve bone mineral values. The study included 37 physically inactive adolescent boys with obesity who were randomly placed into a SIT group (n=18) or a control group (n=19). Participants had to meet inclusion criteria, such as having a BMI at or above the 95th percentile and being healthy and inactive. The SIT group performed 30-second sprints on a cycle ergometer for 12 weeks, three times per week, while the control group maintained their usual lifestyle without any strenuous physical exercise, except mandatory physical education lessons twice weekly. The training intensity during the 12 weeks of SIT was approximately 80-83% of maximum heart rate, and peak power progressively increased from 5.8 to 7.0 W/kg throughout the training period. Fasting blood biochemistry and bone mineral parameters were assessed at baseline and after the 12-week intervention period.

3. Result: Please use the same word (boys or participants) for SIT and CONT group. Otherwise it’s a bit confusing to read.

1. Consider using a table to summarize the training protocol, as this can make it easier for readers to understand the specifics of the SIT program.

2. the authors could discuss the clinical significance of the observed changes in bone mineral and biochemical parameters, and whether these changes are likely to have a meaningful impact on the health outcomes of adolescent boys with obesity.

3. the authors could provide a more detailed explanation of the correlations observed between Δosteocalcin, Δlipocalin-2, and ΔBMI, and discuss the potential implications of these findings for future research.

4. Discuss the implications of the study's findings for clinical practice or public health. For example, could the use of SIT be recommended for obese adolescents who are at risk of osteoporosis or cardiovascular disease?

5. Address any limitations of the study and suggest areas for future research. For example, was the sample size large enough to draw firm conclusions, or were there any confounding variables that might have affected the results?

Additionally, there are some grammar/format suggestions below:

Table 2, please draw a black bottom line at the table.

47 SIT is sprint interval trainings or sprint interval training?

54 with relatively higher

74 is also involved

127 environment

160 subtracting

170 significant level

224 were---was

234 is supported with---by

235 where a negative

274 a significantly higher

294 been—be,

303 from—of,

310 a new insight into how,

Round 2

Reviewer 2 Report (New Reviewer)

Dear authors,

There have been major improvements. In the attachment, I have provided comments and suggested changes regarding the grammar. 

Your paper is on sprint interval training. The warmup is a significant component of the training in your study ranging from 33% to 25%. Therefore you must provide a description of the warmup since it is such a significant component of the training time. 

Kindest regards

Round 3

Reviewer 2 Report (New Reviewer)

Dear authors,

Thank you for the improvements to the paper. I do ask that you check and ensure that the first reference is correct "World Health Organization Obesity and Overweight; 2021;". If it is an online document, you need to cite the date it was accessed and the web link. 

Kindest regards

This manuscript is a resubmission of an earlier submission. The following is a list of the peer review reports and author responses from that submission.

Round 1

Reviewer 1 Report

Interesting topic but this paper could be similar than

J Pediatr Endocrinol Metab   2021 Jun 14;34(8):979-985.  doi: 10.1515/jpem-2021-0216. Print 2021 Aug 26.

Increased lipocalin 2 levels in adolescents with type 2 diabetes mellitus

Could you improve number of cases?

A weak is <50 so no parametric stadistic wull be used, change this

Reviewer 2 Report

I would like to thank the authros for their valuable contribution in this field of research. It is a well written manuscript. However, there are some minor comments that you need to deal with.

Below you will find my minor comments

Abstract 

L19: Please insert "The aim of the current study was". 

Introduction 

L81-82: Please insert the aim of the study prior to your main hypothesis. 

Materials and Methods 

I think that it is well written. 

Results 

Table 1: Please chnage the sympol "@" and try to use plus and minus for standard deviation. 

Reviewer 3 Report

The dataset was already published in this paper: Effect of Sprint Interval Training on Cardiometabolic Biomarkers and Adipokine Levels in Adolescent Boys with Obesity. Though the focus is not the same, but the body composition paragraph rises the concern of salami slicing. Additionally, half of the conclusions of this paper (line 285-288) was published on the previous paper making the current paper not informative enough. The novel part of this research is how the SIT regulates bone-derived biochemical parameters among adolescent boys with obesity. However, they are missing the important factor: diet.

Diet plays an important role in bone-derived biochemical parameters (from reference 27 in this paper). They provide evidence that increased dairy product intake is associated with beneficial changes in circulating levels of bone-related biochemical markers in these girls undergoing a 12-week lifestyle (nutrition counseling and exercise training) intervention program.

There is a significant increase in the height of both groups compared to baseline with unchanged body mass. However, the BMI remains unchanged. How to explain that?

Line 72: No need to refer No 18 again since line 70 already did.

Line 79 should be to the best of our knowledge.
Line 159-160: lacking the verb. Please rewrite the sentence.

Please note what the LL BMD is under Table 1;

Reviewer 4 Report

Comments
The current study investigated effects of sprint interval training (SIT) on adolescent
boys with obesity, with a particular interest in osteokine responses including
osteocalcin, sclerostin, and lipocalin-2. The experiment was designed in the form of a
randomized controlled trial. The primary finding of the current study identified
significant correlations between BMI and bone-derived factors. In addition, the SIT
protocol induced improvement in body composition, anthropometry, and bone
measures.
The main contribution involves the cohort which has not been thoroughly examined
given that potential effects of SIT on adolescent boys with obesity remain largely
unknown. While this study has merits to expand current knowledge base, some
limitations should give rise to cautions when interpreting the results. Please see the
following comments for details.

Introduction
1. SIT needs to be elaborated in the first paragraph. The authors lead the discussion to
SIT by starting with MICT and HIIT which have shown benefits for individuals with
obesity. It would help if comprehensive examinations could be provided as to key
features of SIT that substantiate rationales of applying the training modality to
adolescent boys with obesity. For example, the authors mentioned “all-out” SIT
approach. It is interesting to know the criteria (e.g., heart rate or energy expenditure)
for the “all-out” status in previous studies.
2. Lines 79-83: Please clarify the beneficial changes in bone derived factors
associated reduced body fat in the participants. An explanation on the directions of
beneficial change in osteocalcin, sclerostin, and lipocalin-2, according to the results of
the studies cited in the above paragraph, helps to enhance logic and coherence of the
argument.

Materials and Methods
1. Overall, the authors provided detailed descriptions of the study design and
procedures. The main concern lies in the real time monitor during the exercise. Has
any measurement been conducted (e.g., heart rate, lactic acid concentration, rate of
perceived exertion) to evaluate intensity during SIT?
2. The authors may run a power analysis (by G*power maybe) based on the sample
size.
3. Line 143: I assume the effect size used in statistical analysis is Cohen’s d, please
clarify.
4. Line 151: I agree with using baseline values as covariates in consideration of large
variability among adolescents. I suggest the authors check the statistical method of
other studies because a number of studies often use ANOVA which does not control

baseline performance. I wonder whether the difference in statistical analysis leads to
the non-significant results in the bone derived factors.

Results and Discussion
I will give my comments by combining both sections because a few statements and
conclusions derived from the results need to be revised for more accurate expression.
1. Statistical analysis found no interaction effect in terms of any measures. It is not
very convincing to reach the conclusion that SIT protocol improved bone health in
adolescent boys with obesity. It is difficult to decide whether the improvement is
attributed to pubertal growth or SIT. At this time, the evidence and discussion are not
adequate to fully substantiate the conclusion.
2. Lines 185-186: The authors interpreted the results in a way that change in
osteocalcin was more favorable in the SIT compared to the control. However, the
statement can be misleading because the decreased osteocalcin is interpreted as
favorable change. As the authors mentioned earlier, a reduction of body fat after
supervised MICT increased the level of osteocalcin (lines 67-68).
The data indicated a smaller reduction of osteocalcin in the SIT group than that in the
control group. Please double check whether favorable change here (lines 185-186) is
appropriate.
3. Lines 203-204: Again, if none of the osteokine measures showed favorable change
after SIT, it makes me wonder how to reach the conclusion regarding improvement in
bone health.
4. The authors considered training modality and individual differences of the
participants possible reasons for the inconsistent findings across the existing research.
Maybe it is also important to take a further look at statistical methods given that the
current study took baseline measures as covariates.
5. As for the limitations, my primary concern lies in the relatively high attrition rate of
24% (9 out of 37), which may raise the bias of selection. The authors may add this
point in the revision.

Minor revisions may need in the following places:
Lines 19-21: The sentence is incomplete (“To determine ... in adolescent boys with
obesity”).
Line 168: Typo.
Line 70, 204, 224, and 231: Corrections may need for the in-text citations

Reviewer 5 Report

This study was to analyze the effect of 12-week sprint interval training on bone-related variables in obese male boys.

The topic of this study is interesting, the subjects are appropriate, and there are enough subjects.

However, statistical methods, results, and discussions about them do not attract attention.

The results of the detailed review are as follows.

1. Title- A title that attracts the interest of this study and contains specific contents is needed.

2. Analysis method - Why did this study not conduct two-way repeated measure ANOVA analysis? In order to analyze the effect of exercise in these studies, most of the two-way repeated measure ANOVA is conducted. Therefore, I recommend this analysis method.

3. Unlike previously known results, the results of the two variables (osteocalcin and sclerostin) for the effect of exercise are opposite. Why did this result? As far as I know, osteocalcin increases with exercise.

Wasn't the hypothesis and purpose of your study also intended to confirm that the sprint interval movement increases osteocalcin?

After all, the result of this paper is that there is a negative correlation between osteocalcin and BMI in obese children. I want to ask what this means?

This requires a clear explanation and insight.

4. Only one table is written in this paper. Please check the journal's table preparation regulations and add a table.

5. The results of the reference paper used for reference in the discussion are different. Check the reference paper again.

line 211- 213

"Accordingly, 8-week MICT exercise program showed a significant decrease in undercarboxylated osteocalcin level in young obese males, whereas total osteocalcin 212 level showed downward trend [4]."

Reference paper

[4]. Kim Y-S, Nam JS, Yeo D-W, et al. The effects of aerobic exercise training on serum osteocalcin, adipocytokines and insulin resistance on obese young males. Clin Endocrinol (Oxf) 2015; 82: 686–694. doi:10.1111/cen.12601

Round 2

Reviewer 1 Report

This a simiar article on

Clinical Trial

PLoS One 

. 2017 Aug 14;12(8):e0182970.  doi: 10.1371/journal.pone.0182970.  eCollection 2017.  

Effects of short-term dry immersion on bone remodeling markers, insulin and adipokines 

Marie-Thérèse Linossier 1 , Liubov E Amirova 2  3 , Mireille Thomas 1 , Myriam Normand 1 , Marie-Pierre Bareille 4 , Guillemette Gauquelin-Koch 5 , Arnaud Beck 4 , Marie-Claude Costes-Salon 4 , Christine Bonneau 6 , Claude Gharib 7 , Marc-Antoine Custaud 2  8 , Laurence Vico 1   

Affiliations Expand   

PMID:  28806419  

 PMCID:  PMC5555617   

 DOI:  10.1371/journal.pone.0182970   

 Free PMC article 

Reviewer 3 Report

The revised verson is very informative and address my concerns very well.  The data is presented professionally and clearly. 

Reviewer 4 Report

The revised manuscript is eligible for publication.

Reviewer 5 Report

The authors faithfully explained what was pointed out and I acknowledge their efforts.

They elucidated my concerns about this paper. However, so many changes have been made in English grammar that a detailed review of them will have to be made.

For example, double check the spelling of line 106 (training in) word. Overall, please conduct a complete error review of the paper once again.